# Taste Preferences and Orosensory Feed Testing Behavior in Barramundi *Lates calcarifer* (Latidae, Perciformes)

Alexander Kasumyan [1,2,*] , Olga Isaeva [3] and Le T. K. Oanh [4]

1   Department of Ichthyology, Faculty of Biology, Lomonosov Moscow State University, Leninskiye Gory 1, Page 12, 119234 Moscow, Russia

2   Russia A.N. Severtsov Institute of Ecology and Evolution, Leninskiy pr. 33, 119071 Moscow, Russia

3   Department of Water Bioresources and Aquaculture, Kamchatka State Technical University, Vilyuiskaya st. 56, 683001 Petropavlovsk-Kamchatsky, Russia

4   Coastal Branch, Joint Vietnam-Russia Tropical Science and Technology Research Center, Nguyen Thien Thuat st. 30, Nha Trang 57127, Khanh Hoa, Vietnam

*   Correspondence: alex_kasumyan@mail.ru; Tel.: +7-495-939-27-25 or +7-985-987-78-99

**Abstract:** In the cultivation of barramundi *Lates calcarifer*, one of the key factors is developing new commercial feeds that are nutritionally adequate and attractive to fish. The sensory quality of feeds can be improved by taste stimulants. The aim of the present study is to examine the taste attractiveness of 21 free L-amino acids and 4 basic taste substances (0.1–0.001 M). The feeding behavior that accompanied the orosensory testing of feed is also studied. Agar pellets flavored with each taste substance were individually offered to barramundi juveniles (5–9 cm, 4–10 g). Citric acid, cysteine, and alanine were palatable; sucrose and 7 amino acids had an aversive taste; sodium chloride, calcium chloride, and 12 amino acids did not influence the consumption of pellets. Taste preferences for amino acids are positively correlated in 6 out of 21 fish species, which confirms high species specificity of taste preferences in fishes. Barramundi often make repeated grasps and rejections of pellets regardless of their palatability, which led to the destruction of pellets in up to 50% of trials. When swallowing, fish retain pellets several times longer, and pellet fragmentation occurs more often, than in the case of final refusal of pellets. The data presented here can be used to improve the cultivation of barramundi.

**Keywords:** amino acids; citric acid; taste attractiveness; feeding behavior; taste stimulants

## 1. Introduction

Barramundi *Lates calcarifer*, also known as seabass in Asia, is a euryhaline catadromous fish widely distributed across coastal waters of the Indo-Pacific region, from the Arabian Gulf to China, Papua New Guinea, and northern Australia [1]. Barramundi is an opportunistic predator, mainly feeding on macrocrustaceans and pelagic teleost fish. It can grow up to 2 m in length and weighs more than 60 kg [2]. Owing to its high flesh quality and fast growth rate, barramundi have become one of the most important aquaculture finfish in many countries of Southeast Asia and Australia, with global production of farmed barramundi reaching 75,000 metric tons annually [3–6].

In the cultivation of barramundi, as well as the cultivation of other farmed fish, one key factor involves the finding of new and less expensive commercial feeds that will be nutritionally adequate and contain a sensory quality is palatable to the fish. Applying such feeds into aquaculture practice should reduce the cost of farmed fish production, as well as decrease nutrient loss and food waste, and, consequently, diminish degradation of local environments [7,8]. The sensory quality of feeds can be substantially improved using feeding stimulants perceived by the fish's gustatory system, a remarkably well-developed system when compared to other groups of vertebrates [9]. Many fish species possess an exceptionally high number of taste buds, which are spread not only in the oral cavity but also over the head and trunk surfaces, and can even be found in fins, including the

caudal fin [10,11]. Numerous electrophysiological studies show that fish's sensitivity to taste stimuli is significantly higher than in other vertebrates, such as amphibians and mammals [12].

Taste preferences in fishes are highly diverse and species-specific, and do not depend on fish phylogeny, systematics or trophic category, and individual feeding experience. The same substance, when tasted by different fish species, can stimulate swallowing in one and a rejection of the food by another [13]. Being genetically determined, taste preferences remain stable over generations and do not shift under various environmental conditions [14]. Basic taste substances that would evoke sweet, sour, salty, and bitter sensations in humans, such as carboxylic and other organic acids, sugars, and various low-molecular nitrogenous substances, have been evaluated for taste attractiveness for many fish species [13]. In most cases, free amino acids, which are diverse and present in all prey organisms, are used as the taste stimuli for both fundamental studies on fish taste perception and in the search for taste stimulants for farmed fish species [9]. It has been found that many amino acids are highly palatable to fish and can improve food intake when added into feeds [15–19].

Recently, the taste attractiveness of several echinoderm species, including the pest crown-of-thorns sea star *Acanthaster planci*, was examined as a feed source for barramundi juveniles [20]. Barramundi was also chosen as one of the model fish for assessing the taste attractiveness of aposematic and cryptic colored marine gastropods in the Ovulidae (Caenogastropoda) family [21]. Nevertheless, data concerning taste preferences of barramundi in regard to any defined substances are absent, to the best of our knowledge. The aim of our study is to examine the taste attractiveness of free proteinogenous L-amino acids and basic taste substances in barramundi juveniles. We also intend to study the feeding behavior that accompanies oral evaluation of feed preferences in barramundi. Our findings reveal that taste preferences in barramundi are specific and do not correlate with taste preferences of many other fish species. The spectrum of taste substances with taste attractivity is narrow and includes 3 out of 25 substances tested. In contrast, the behavior of barramundi when orosensory tested feeds is similar to the feeding behavior of other fish species.

## 2. Materials and Methods

### 2.1. Fish and Maintenance

The trials were conducted in facilities at the Coastal Branch of the Joint Vietnam–Russia Tropical Science and Technology Research Center (Nha Trang City, Vietnam). Juvenile barramundi, *Lates calcarifer* (72 naïve individuals), were used in trials. Barramundi were obtained from local fish farms (Công ty Trách nhiệm Hữu hạn Thủy sản Khánh Hòa) and were transported in aerated tanks to the laboratory. In the first week, fish were kept in common aquariums (20 fish per 150–250 L) equipped with an internal filter submersible water pump (AP350) and an air pump (P-125). The water temperature ranged from 28 to 30 °C. Every other day, half of the water in the tank was replaced with clean sea water. During this period, the fish were fed ad libitum with crushed fresh shrimp, *Penaeus vannamei*, once a day.

A few days before a series of trials started, randomly selected fish were placed in individual aquariums ($29 \times 25 \times 25$ cm; 15 L) with opaque lateral walls, which prevented visual contact with neighboring fish. The aquariums had no gravel and were equipped with an aperture in the cover for feeding and introducing experimental pellets. Aquariums were combined into two closed systems. In each system, water circulated through an external aerated biofilter ($60 \times 45 \times 80$ cm; 200 L) and was pumped into all 12 aquariums simultaneously (1.5–1.7 L/min) using an AP5600 pump (Guangdong Zhenhhua Electrical Appliance Co., Ltd.; Zhongshan, China). The water temperature and water salinity ranged from 28 to 30 °C and from 32 to 33%, respectively. Every 3–5 days, half of the water circulating in the systems was replaced with fresh sea water. Fish were fed to satiation every day at 18:00 (local time) with fresh or fresh–frozen crushed shrimps.

Trials were performed in three successive series in April–May 2018. Series 1 tested 4 classical taste substances that evoke basic taste sensations in humans: sweet (sucrose),

sour (citric acid), bitter (calcium chloride), and salt (sodium chloride). Series 2 tested 21 L-amino acids (henceforth, all amino acids mentioned are in L-form except glycine, which does not have L- or D-isomers). Series 3 tested citric acid at different concentrations. For each series, 24 new fish were used: in Series 1 and 2, individuals of 5–7 cm in total length and 4–7 g of body weight were used, while in Series 3 individuals of 6–9 cm and 5–10 g were used for testing.

### 2.2. Preparation of Pellets

A hot (60–70 °C) agar solution (2%, Reanal) was mixed with a red dye solution (Ponceau 4R) and with a solution of one of the taste substances, or else with a supernatant of shrimp, *P. vannamei,* water extract. To prepare the extract, fresh shrimp were homogenized in a ceramic mortar, and after a 15 min extraction using the appropriate amount of water at 20 °C, the homogenate was centrifuged at 4000 rpm for 20 min at +3 °C (Eppendorf Centrifuge 5430R). The concentration of Ponceau 4R in agar matrix was 5 μM; the concentration of shrimp water extract was 300 mg (wet weight)/mL. The majority of substances used in this study were tested at a concentration of 100 mM except glutamic acid, aspartic acid, leucine, isoleucine, tryptophan (10 mM), and tyrosine (1 mM) owing to their low solubility in water. Blank agar gel (the control pellets) only contained the dye. Sea water was used to prepare all solutions and extracts.

The hot mixture was poured into a Petri dish and cooled agar gels were stored at +5 °C for up to 2 weeks, except for the agar gel with shrimp water extract, which was stored for no more than 1–2 days. Cylindrical pellets were cut from the agar gel with a thin-walled stainless-steel tube just before each trial. All pellets had a length of 4.0 mm and a diameter of 1.15 mm in Series 1 and 2, and that of 2.0 mm in Series 3.

### 2.3. Experimental Procedure

For 2–3 days before the trials, the fish were trained to grasp individually offered agar pellets containing a water extract of shrimp, *P. vannamei*. Trained fish grasped a pellet within 3–5 s after it was dropped into the water. This practice prevented the diffusion of a water-soluble substances from the pellet [22]. In each trial, one pellet was introduced into the aquarium, while several characteristics of the fishes' response were recorded: (i) the number of grasps of the pellet, (ii) the retention time of the pellet in the fish's mouth during the first grasp, (iii) the retention time during all grasps, and (iv) the ingestion of or refusal to consume the pellet at the end of the trial. Fragmentation of the pellet into pieces (which the fish thereafter either swallowed or ignored) was recorded as an additional characteristic. In some experiments with fragmentation, fish consumed only half of the pellet, which was also considered.

A grasped pellet inside of the mouth triggered easy-to-see chewing movements, as well as more frequent and higher amplitude movements of gill covers (operculum). Abrupt restoration back to a normal breathing rate and cessation of chewing movements indicated the moment when the fish had swallowed the pellet. A fish's refusal to consume a rejected pellet was registered if the fish lost all the interest in the pellet and swam away from it to another part of the aquarium. In this case, the trial was considered complete; trials were also considered complete after ingestion. If the pellet was not grasped within 1–2 min after introduction, or consumption of the pellet could not be determined because of fragmentation, the trial was disregarded. The rejected pellet or its fragments were removed from the aquarium immediately after the end of the trial.

Trials were performed from 10:00 to 15:00 for about six weeks. Several control pellets and pellets containing taste substances were presented to each fish in a random order at 15–20 min intervals. The same number of repetitions was performed for each extract on each fish. Pellets containing a shrimp extract were used to verify the motivation of fish to feed. All measurements were made by the same observer. This procedure minimized any bias related to the fish's potential exhaustion or feeding motivation and did not evoke confounding effects.

*2.4. Statistical Analysis*

In total, 3660 trials were conducted, including 840, 2976, and 216 trials for Series 1, 2, and 3. The data were presented as mean ± standard error of mean (s.e.m.).

Chi-squared tests were conducted to detect significant differences between the consumption of flavored and control pellets. The Mann–Whitney U-test was used to compare differences in all other characteristics of fish behavior. The non-parametric Spearman rank correlation coefficient ($r_s$) was also used to evaluate the relationship between taste responses of different fish species for the same range of substances, and between variables of fish feeding behavior. Statistical analyses were carried out with STATISTICA (version 10, StatSoft) and Stadia-8.0 (SPA "Informatics and computers").

Differences were considered significant when the *p*-value was $p < 0.05$ (*), $p < 0.01$ (**), and $p < 0.001$ (***). A *p*-value greater than 0.05 was considered not significantly different.

*2.5. Ethical Statement*

The authors confirm that all experiments were performed in accordance with relevant guidelines and regulations without causing pain, suffering, distress, lasting harm, or death to fish involved in research. In addition, the Joint Vietnam–Russia Tropical Science and Technology Research Center approved the experimental protocol used in the present study (permit # 856/QD-TTNDVN).

## 3. Results

*3.1. Background Behavior*

Our barramundi test group easily tolerated the transfer to individual aquariums. Within a few minutes fish began to swim calmly, and a researcher approaching the aquarium did not cause behavior suggesting fear, such as hiding at the bottom or jumping out of the water, which is a typical reaction for many other fish species after a new aquarium transfer. The normal behavior of barramundi is characterized by slow movements in the aquarium, mainly near the place of food supply, interrupted by period of freezing in the water column, usually head down at about 20–50° (Figure 1). Already, on the first day after the transfer, the fish began to grasp pieces of fresh shrimp dropped into the water or held with tweezers. Fish also readily grasped agar pellets with shrimp extract offered for training.

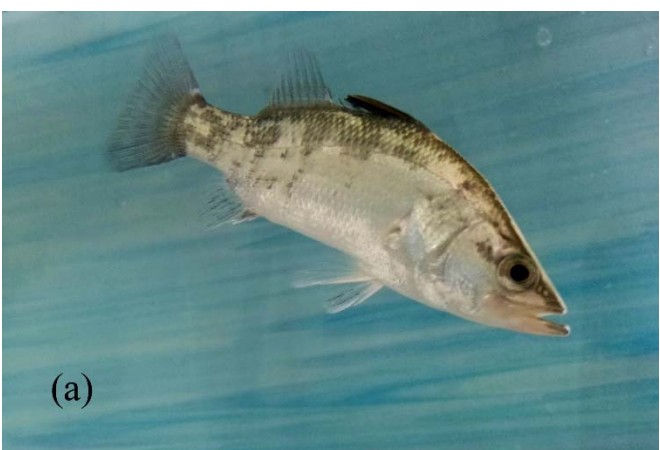

(a)

**Figure 1.** *Cont.*

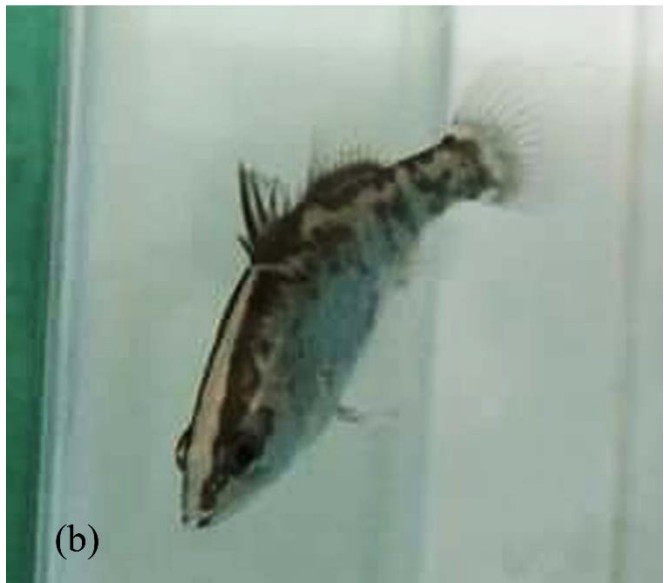

**Figure 1.** Barramundi *Lates calcarifer* juvenile in the typical waiting pose in an experimental aquarium: profile (**a**) and front (**b**) view.

### 3.2. Taste Attractiveness of Basic Taste Substances

Only citric acid was found palatable among the four substances that evoke basic taste sensations in humans (Series 1). The efficiency of citric acid was close to that of the shrimp water extract. Both calcium chloride and sodium chloride were ineffective taste substances and did not influence the consumption of pellets, while pellets flavored with sucrose were consumed more than twice as infrequently in relation to the control (Figure 2).

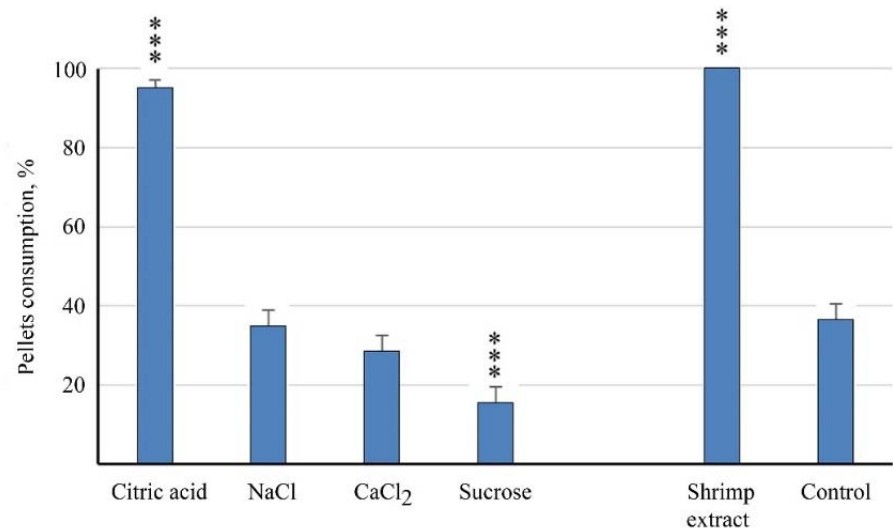

**Figure 2.** Palatability of agar gel (2%) pellets flavored with basic taste substances, 0.1 M, and shrimp *Penaeus vannamei* water extract, 300 g (wet weight)/L for barramundi *Lates calcarifer* juveniles. ***—differences from the control are significant at $p < 0.001$ ($\chi^2$-test). Whiskers are the s.e.m.

### 3.3. Taste Attractiveness of Amino Acids

Among the 21 free amino acids, cysteine proved the most palatable (Series 2). The pellets flavored with cysteine were consumed nearly 3 times more than the control pellets, though less often than the pellets containing shrimp water extract. Pellets with citric acid, used in this series as an additional positive control, were consumed with the same willingness as pellets flavored with cysteine. Alanine also significantly stimulated pellet

consumption, but the effect was not as strong as cysteine. Seven amino acids had an aversive taste; their presence in pellets led to a considerable reduction in consumption. The most pronounced negative effect was observed in regard to pellets containing tyrosine, which reduced consumption by nearly 13-fold in relation to the control. Twelve amino acids did not influence the consumption of pellets (Figure 3).

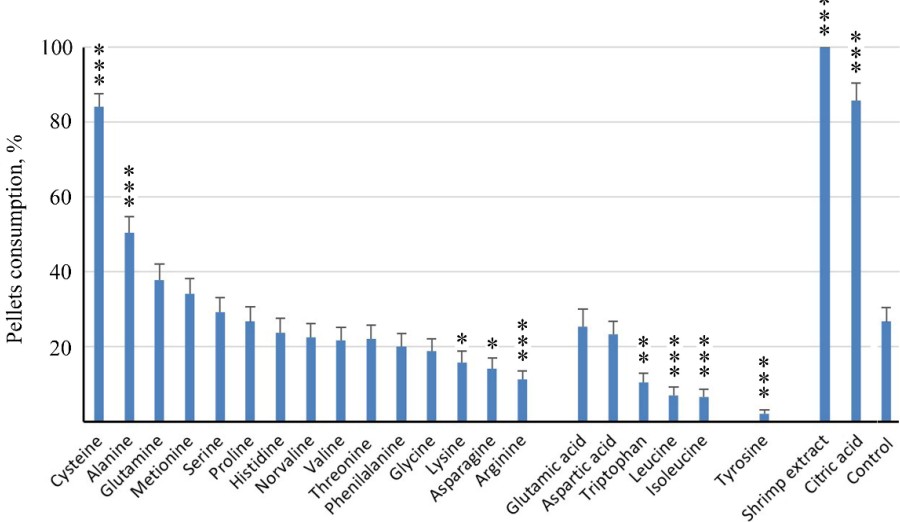

**Figure 3.** Palatability of agar gel (2%) pellets flavored with L-amino acids and shrimp *Penaeus vannamei* water extract, 300 g (wet weight)/L for barramundi *Lates calcarifer* juveniles. The concentrations are 0.1 M for all substances except glutamic acid, aspartic acid, leucine, isoleucine and tryptophan (0.01 M), and tyrosine (0.001 M). *, **, and ***—differences from the control are significant at $p < 0.05$, $p < 0.01$, and $p < 0.001$, respectively ($\chi^2$-test). Whiskers are the s.e.m.

### 3.4. Citric Acid

Citric acid was chosen for the trials that aimed to evaluate the concentration of the taste substance needed in order to produce sufficient palatability (Series 3). Barramundi consumed pellets flavored with 0.1 M citric acid almost to the same degree that they consumed such pellets in Series 1 and 2. A 10-fold decrease in the concentration of citric acid led to a 2.2-fold decrease in the consumption of pellets; however, the difference in relation to the control was still significant (Figure 4).

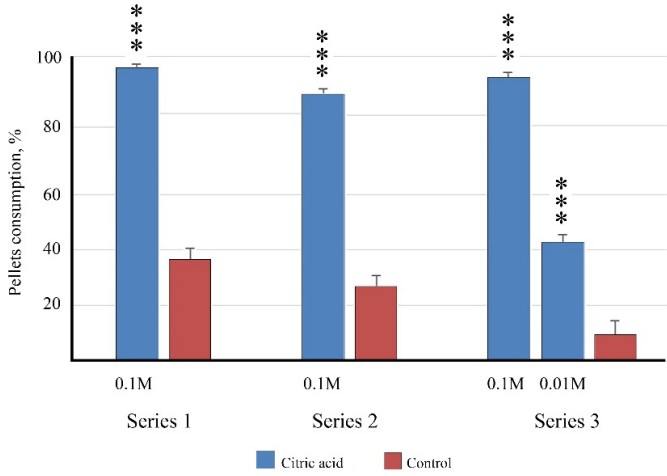

**Figure 4.** Palatability of agar gel (2%) pellets flavored with citric acid for barramundi *Lates calcarifer* juveniles in different series of trials. ***—differences from the control are significant at $p < 0.001$ ($\chi^2$-test). Whiskers are the s.e.m.

### 3.5. Feeding Behavior

Trained barramundi usually grasped any offered pellet within the first 3–5 s at about 3–5 cm from the surface of the water. To do this, the fish visually detect a pellet that had fallen into the water, approach it, and then rushed quickly to grasp it. We rarely observed the barramundi grasp pellets lying at the bottom of the tanks.

Often barramundi rejected the grasped pellet before re-grasping it. The maximum number of grasps reached 22 in a trial where the pellet contained norvaline. Nevertheless, in most trials, one or two grasps were observed regardless of the palatability of the pellets (Table 1; Figure 5). Such manipulations often led to the destruction of the pellet and the appearance of several fragments, which were often re-grasped. The percentage of trials with pellet destruction did not exceed 50% and occurred least often in trials involving highly palatable cysteine and citric acid. Moreover, fish did not destroy a single pellet with shrimp extract in both Series 1 and 2 (Figure 6). However, no significant correlation was found between the consumption of pellets with amino acids and the number of grasps and fragmentation (Figure 7). Pellets could be fragmented even in trials where a single grasp occurred and the pellet retention time was less than one second.

**Table 1.** Behavioral taste responses of barramundi *Lates calcarifer* to pellets of agar gel (2%) flavored with 4 basic taste substances, 21 L-amino acids and shrimp *Penaeus vannamei* water extract. *, **, and ***—differences from the control are significant at $p < 0.05$, $p < 0.01$, and $p < 0.001$, respectively (Mann–Whitney U-test).

| Substance | Number of Pellet Grasps over the Entire Trial | Pellet Retention Time, s | | Number of Trials |
|---|---|---|---|---|
| | | After the First Grasp | Over the Entire Trial | |
| Series 1: basic taste substances | | | | |
| Citric acid, 0.1 M | 1.3 ± 0.1 *** | 9.6 ± 0.4 *** | 10.8 ± 0.4 * | 120 |
| Sodium chloride, 0.1 M | 3.8 ± 0.2 | 3.6 ± 0.3 | 12.3 ± 0.6 | 120 |
| Calcium chloride, 0.1 M | 4.1 ± 0.2 | 3.2 ± 0.3 ** | 12.3 ± 0.6 | 120 |
| Sucrose, 0.1 M | 4.7 ± 0.2 ** | 2.7 ± 0.2 *** | 12.4 ± 0.5 | 120 |
| Shrimp water extract,300 g (wet weight)/L | 1.1 ± 0.0 *** | 9.7 ± 0.4 *** | 10.5 ± 0.4 * | 240 |
| Control | 3.7 ± 0.2 | 4.2 ± 0.3 | 12.2 ± 0.5 | 120 |
| Series 2: amino acids | | | | |
| Cysteine, 0.1 M | 1.2 ± 0.1 *** | 7.5 ± 0.4 *** | 8.2 ± 0.3 | 120 |
| Alanine, 0.1 M | 1.8 ± 0.1 ** | 7.6 ± 0.6 *** | 12.1 ± 0.8 *** | 120 |
| Glutamine, 0.1 M | 1.8 ± 0.1 *** | 5.0 ± 0.4 *** | 8.2 ± 0.6 | 120 |
| Methionine, 0.1 M | 1.5 ± 0.1 *** | 5.3 ± 0.5 *** | 8.0 ± 0.6 | 120 |
| Serine, 0.1 M | 1.5 ± 0.1 *** | 4.0 ± 0.4 *** | 6.0 ± 0.6 ** | 120 |
| Proline, 0.1 M | 2.3 ± 0.1 ** | 3.8 ± 0.3 *** | 8.9 ± 0.8 | 120 |
| Histidine, 0.1 M | 2.4 ± 0.2 | 3.5 ± 0.4 | 8.1 ± 0.6 | 120 |
| Norvaline, 0.1 M | 1.9 ± 0.2 ** | 3.5 ± 0.4 | 6.8 ± 0.8 * | 120 |
| Valine, 0.1 M | 1.6 ± 0.1 *** | 3.3 ± 0.3 * | 5.3 ± 0.4 *** | 120 |
| Threonine, 0.1 M | 1.5 ± 0.1 *** | 3.7 ± 0.4 * | 5.7 ± 0.5 ** | 120 |
| Phenylalanine, 0.1 M | 1.6 ± 0.1 *** | 2.9 ± 0.3 | 5.1 ± 0.4 *** | 120 |
| Glycine, 0.1 M | 1.9 ± 0.1 ** | 3.3 ± 0.3 *** | 6.5 ± 0.5 | 120 |
| Lysine, 0.1 M | 1.6 ± 0.1 *** | 3.4 ± 0.4 | 5.7 ± 0.5 ** | 120 |
| Asparagine, 0.1 M | 2.1 ± 0.2 | 2.7 ± 0.3 | 6.2 ± 0.5 * | 120 |
| Arginine, 0.1 M | 1.9 ± 0.1 * | 2.7 ± 0.3 | 5.8 ± 0.5 ** | 120 |
| Glutamic acid, 0.01 M | 1.9 ± 0.1 * | 3.1 ± 0.3 * | 6.7 ± 0.5 | 120 |
| Aspartic acid, 0.01 M | 1.8 ± 0.1 *** | 3.7 ± 0.3 *** | 6.6 ± 0.5 | 120 |
| Tryptophan, 0.01 M | 1.7 ± 0.1 ** | 2.2 ± 0.2 | 5.0 ± 0.4 *** | 120 |
| Leucine, 0.01 M | 1.8 ± 0.1 *** | 2.1 ± 0.2 | 4.5 ± 0.5 *** | 120 |
| Isoleucine, 0.01 M | 1.6 ± 0.1 *** | 1.9 ± 0.2 | 3.8 ± 0.4 *** | 120 |
| Tyrosine, 0.001 M | 1.6 ± 0.1 *** | 1.8 ± 0.1 | 4.1 ± 0.4 *** | 120 |
| Shrimp water extract,300 g (wet weight)/L | 1.0 ± 0.0 *** | 8.8 ± 0.3 *** | 8.9 ± 0.3 | 240 |
| Citric acid, 0.1 M | 1.1 ± 0.0 *** | 9.6 ± 0.7 *** | 10.6 ± 0.6 ** | 96 |
| Control | 2.5 ± 0.2 | 2.9 ± 0.3 | 9.0 ± 0.8 | 120 |
| Series 3: citric acid | | | | |
| Citric acid, 0.1 M | 1.7 ± 0.2 *** | 14.5 ± 0.9 *** | 17.0 ± 1.0 *** | 72 |
| Citric acid, 0.01 M | 1.8 ± 0.2 ** | 6.3 ± 0.7 *** | 8.9 ± 0.8 | 72 |
| Control | 2.8 ± 0.3 | 2.4 ± 0.3 | 8.3 ± 1.0 | 72 |

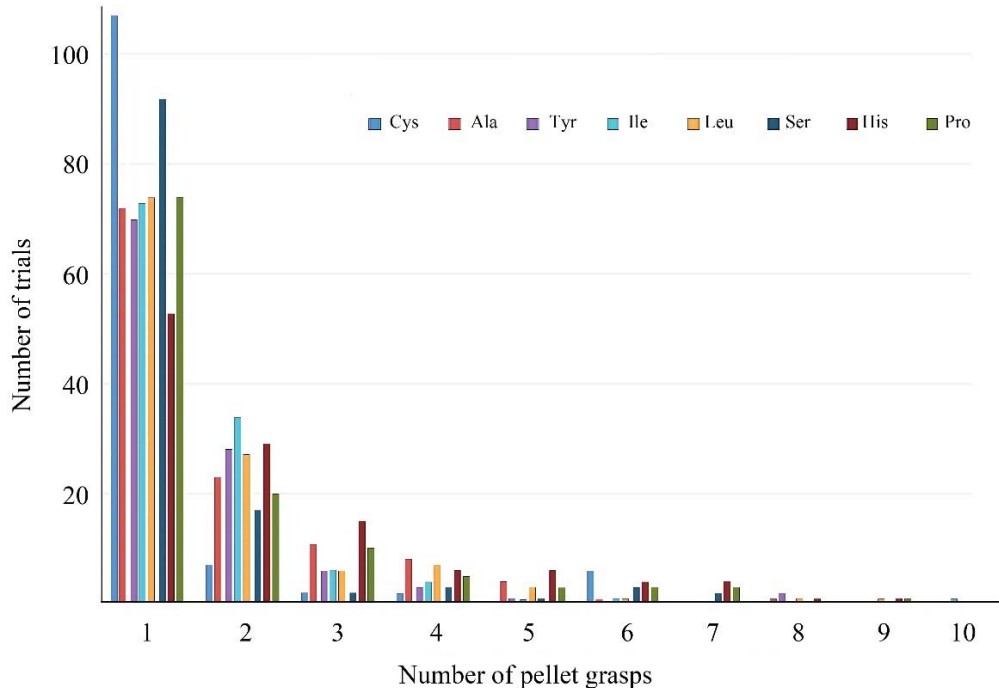

**Figure 5.** Frequency distribution of trials in relation to number of grasps per trial with pellets flavored with amino acids different by taste attractiveness for barramundi *Lates calcarifer* juveniles. Palatable amino acids: 1—cysteine (0.1 M) and 2—alanine (0.1 M); amino acids with indifferent taste: 3—serine (0.1 M), 4—histidine (0.1 M), and 5—proline (0.1 M); aversive amino acids: 7—tyrosine (0.001 M), isoleucine (0.01 M), and leucine (0.01 M).

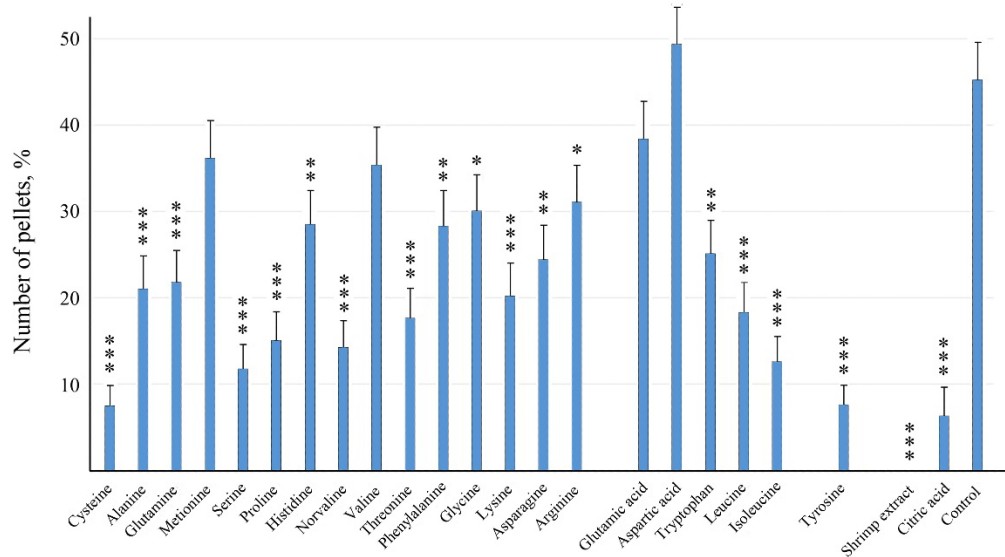

**Figure 6.** Fragmentation of agar gel (2%) pellets flavored with amino acids in barramundi *Lates calcarifer* juveniles. *, ** and ***—differences from the control are significant at $p < 0.05$, $p < 0.01$, and $p < 0.001$, respectively ($\chi^2$-test). Whiskers are the s.e.m.

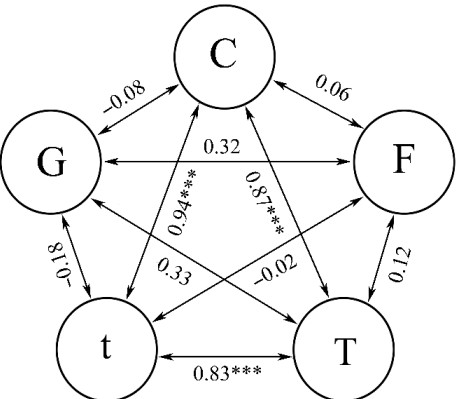

**Figure 7.** Spearman rank correlations between different parameters of behavioral taste responses to pellets of agar gel flavored with amino acids observed in barramundi *Lates calcarifer*. C—pellets consumption; F—pellets fragmentation; G—number of pellet grasps; t and T: pellet retention times for the first grasp and for all grasps over the entire trial, respectively. *** $p < 0.001$.

Fish retained the pellet in the mouth for up to 7–10 s after the first grasp and up to 10–12 s throughout the entire trial. The total retention time reached 25–43 s in some trials with pellets containing citric acid, cysteine, and shrimp extract. The higher the palatability of the pellets, the longer the first and total retention time; this relationship is significantly correlated, as is the relationship between the duration of the first grasp and total retention time (Figure 7).

Trials that ended with the barramundi swallowing the pellet displayed a number of grasps similar to the number of grasps in trials in which the pellet was finally rejected, with the exceptions of trials for cysteine and alanine where the number of grasps was significantly lower when pellets were swallowed, and trials for glutamic acid where the number of grasps was lower when pellets were finally rejected (Figure 8). The pellet retention time for pellets containing amino acids was significantly longer when pellets were swallowed, 3–6 and 1.5–4 times more often during the first grasp and during the sum of all grasps, respectively. (Figures 9 and 10). Fragmentation of most pellets containing amino acids occurred more frequently (though not always) in trials that ended with swallowing of the pellets (Figure 11).

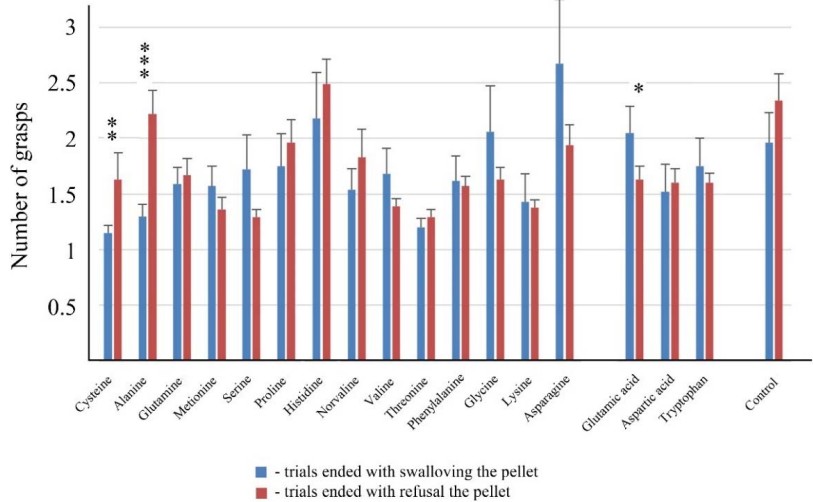

**Figure 8.** The number of grasps of pellets flavored with amino acids ended with swallowing or refusal of the pellet by barramundi *Lates calcarifer*. *, ** and ***—differences between trials performed with each amino acid are significant at $p < 0.05$, $p < 0.01$, and $p < 0.001$, respectively (Mann–Whitney U-test). Whiskers are the s.e.m.

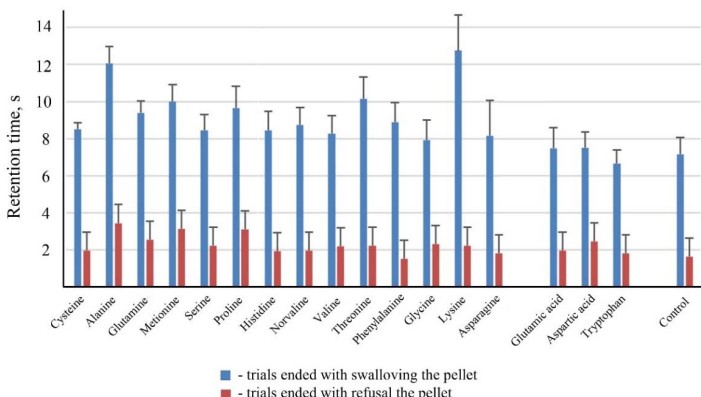

**Figure 9.** Retention time after the first grasp of pellets flavored with amino acids, ended with swallowing or refusal of the pellet by barramundi *Lates calcarifer*. Differences between trials completed with each amino acid are significant at $p < 0.001$ for all substances (Mann–Whitney U-test). Whiskers are the s.e.m.

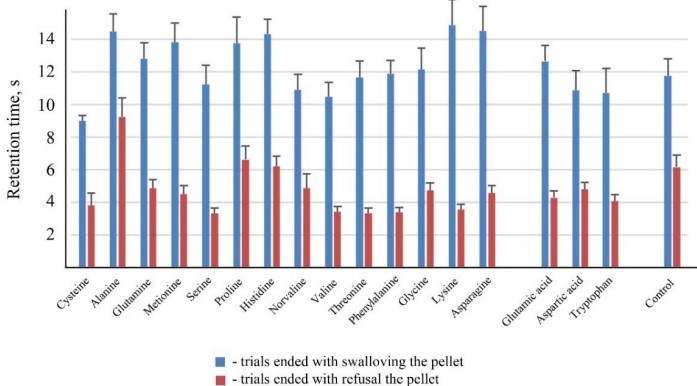

**Figure 10.** Retention time during all grasps of pellets flavored with amino acids ended with swallowing or refusal of the pellet by barramundi *Lates calcarifer*. Differences between trials performed with each amino acid are significant at $p < 0.001$ for all substances (Mann–Whitney U-test). Whiskers are the s.e.m.

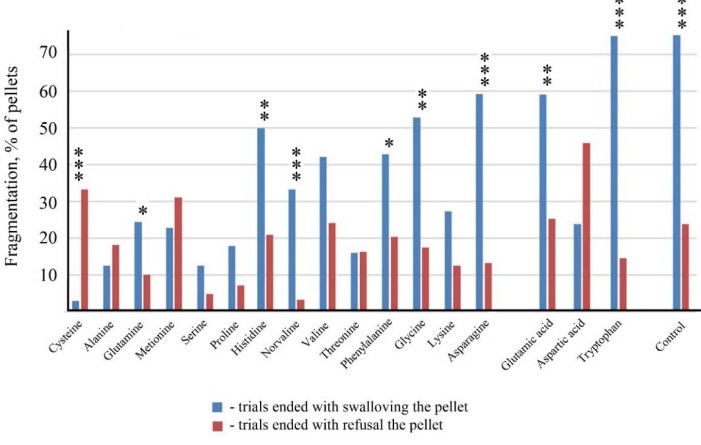

**Figure 11.** Fragmentation of pellets flavored with amino acids ended with swallowing or refusal of the pellet by barramundi *Lates calcarifer*. *, ** and ***— differences between trials ended with swallowing and refusal of the pellet are significant at $p < 0.05$, $p < 0.01$, and $p < 0.001$, respectively ($\chi^2$-test).

## 4. Discussion

In more than three dozen species studied in previous research, taste preferences have been assessed mainly in fish that inhabit water bodies in the temperate (boreal) climate zone [13]. For tropical fish, such data are known only for a few freshwater species, including marble goby, *Oxyeleotris marmoratus*; pearl gourami, *Trichopodus leerii*, and three spot gourami, *T. trichopterus*; Nile tilapia, *Oreochromis niloticus*; and North African catfish, *Clarias gariepinus* [23–28]. For marine tropical fish, taste preferences have been studied for grouper *Epinephelus fuscoguttatus* and the hybrid grouper *E. fuscoguttatus* × *E. lanceolatus* [29,30]. Our study is the first to report the taste preferences in a tropical euryhaline fish species. We found that only 3 substances among the 25 tested have an attractive taste to barramundi, while more substances (8) have a repulsive taste. The remaining 14 substances have a taste that leaves barramundi indifferent.

Only 2 amino acids have an attractive taste to barramundi, cysteine and alanine. These amino acids are more often among those most attractive to different fish: cysteine for 18 tested species and alanine for tested 20 species [31]. As with barramundi, both cysteine and alanine have a taste that is attractive to common carp, *Cyprinus carpio*; tench, *Tinca tinca*; stellate sturgeon, *Acipenser stellatus*; North African catfish; and three spot gourami [23,28,32–34]. However, there are species that refuse to consume pellets if cysteine or alanine is present—these include goldfish, *Carassius auratus*; European minnow, *Phoxinus phoxinus*; Siberian sturgeon, *Acipenser baerii*; chum salmon, *Oncorhynchus keta*; and Arctic char, *Salvelinus alpinus erythrinus* [34–38]. Comparison of barramundi with other fish species in terms of taste preferences for amino acids shows that there is no significant correlation with 14 of the 21 species. Among the 6 species that are positively correlated with barramundi in terms of taste preferences, there are 3 species that are, like barramundi, salinity tolerant fish: Persian sturgeon, *Acipenser persicus*; three-spined stickleback, *Gasterosteus aculeatus*; and nine-spined stickleback, *Pungitius pungitius* (Table 2). It is noteworthy that there is no correlation with other euryhaline fish species, including Russian sturgeon *Acipenser gueldenstaedtii*, stellate sturgeon, chum salmon, and Nile tilapia. For this reason, we cannot consider the similarities in taste preferences for fish that easily tolerate changes in osmotic environments. The results of the comparisons fully confirm the previously formulated general rule, that taste preferences in fishes are highly species-specific [13].

Palatable cysteine and alanine belong to a category of non-essential amino acids, which fish can synthesize. In contrast, 5 out of 7 amino acids with deterrent tastes (arginine, leucine, isoleucine, lysine, and tryptophan) are essential amino acids, and can only be obtained through food. No significant correlation was found between the taste attractiveness of amino acids and their content in newly fertilized barramundi eggs ($r_s = -0.15$; $p > 0.05$). It is noteworthy that a significant correlation was also not found between taste attractiveness of essential amino acids and their content in whole barramundi bodies ($r_s = 0.57$; $p > 0.05$; data on essential amino acid content in newly fertilized eggs and whole barramundi were used from Dayal et al. (2003) [43] and Glencross (2006) [44], respectively). The latter supports the results of meta-analysis found that taste attractiveness is not linked to essential nutrients [31]. For humans, many essential amino acids have a bitter taste, while non-essential amino acids do not evoke aversive taste responses [45–47].

Citric acid also proved highly palatable for barramundi. A strong stimulating effect of citric acid was obtained in each of the 3 series of trials in the present study (Figure 4), as well as in a previous study aimed at evaluating the palatability of various tropical echinoderms [20]. It is commonly believed that detection of sour flavors was the most ancient taste to evolve. Moreover, in contrast to other tastes, the sour taste does not appear to have been lost in any major vertebrate taxa. For most vertebrates, the sour taste is aversive, indicating inappropriate or dangerous food. Animals that enjoy the sour taste triggered by acidic foods are exceptional [48].

**Table 2.** The values of Spearman rank correlation coefficient for taste preferences to 21 free L-amino acids between barramundi *Lates calcarifer* and other fish species.

| Fish Species | $r_s$ | References |
|---|---|---|
| Siberian sturgeon, *Acipenser baerii* [1] | −0.19 | [34] |
| Russian sturgeon, *Acipenser gueldenstaedtii* | 0.10 | [34] |
| Persian sturgeon, *Acipenser persicus* [2] | 0.47 * | [34] |
| Stellate sturgeon, *Acipenser stellatus* | 0.16 | [34] |
| Brown trout, *Salmo trutta caspius* | −0.08 | [39] |
| Chum salmon, *Oncorhynchus keta* | −0.08 | [37] |
| Arctic charr, *Salvelinus alpinus erythrinus* | 0.20 | [38] |
| Common carp, *Cyprinus carpio* | 0.24 | [32] |
| European minnow, *Phoxinus phoxinus* | 0.42 | [36] |
| Roach, *Rutilus rutilus* | −0.06 | [35] |
| Tench, *Tinca tinca* | 0.70 *** | [33] |
| Goldfish, *Carassius auratus* | −0.53 * | [35] |
| Stone loach, *Barbatula barbatula* | 0.17 | [40] |
| North African catfish, *Clarias gariepinus* | 0.85 *** | [23] |
| Nine-spined stickleback, *Pungitius pungitius* [3] | 0.68 *** | [41] |
| Three-spined stickleback, *Gasterosteus aculeatus* [4] | 0.80 *** | [42] |
| Guppy, *Poecilia reticulata* | 0.19 | [35] |
| Pearl gourami, *Trichopodus leerii* | 0.18 | [26] |
| Three spot gourami, *Trichopodus trichopterus* | 0.67 *** | [28] |
| Nile tilapia, *Oreochromis niloticus* | −0.01 | [27] |
| Arctic flounder, *Liopsetta glacialis* | 0.40 | [35] |

[1]—the correlation coefficient was calculated based on the taste responses to 19 free amino acids (with exception cysteine and norvaline); [2]—the correlation coefficient was calculated based on the taste responses to 20 free amino acids (with exception norvaline); [3]—the Moskva River population was chosen for comparison; [4]—the Baltic population was chosen for comparison. Significant value is * $p < 0.05$ and *** $p < 0.001$.

Indeed, for many fish, citric acid has a deterrent taste [13,49,50]. Nevertheless, there are many species of fish for which citric acid is highly palatable and substantially stimulates food intake. This is true not only of barramundi, but also Nile tilapia; redbelly tilapia, *Coptodon zillii*; grass carp, *Ctenopharyngodon idella*; brown trout, *Salmo trutta*; guppy, *Poecilia reticulata*; Arctic char; common carp; tench; marble goby; and three-spined and nine-spined sticklebacks [13,25,27,32,33,35,38,39,41,42,51,52]. Various blends of organic acids, including citric acid, so called dietary acidifiers, are used as additives to improve formulated feed intake of farmed fish [53,54]. Taste attractiveness of different carboxylic acids varies as widely as that of amino acids [51,52,55]. Moreover, some carboxylic acids are even more taste attractive than citric acid [33]. Therefore, further elucidation of the palatability of carboxylic acids is crucial in our search for taste stimulants for barramundi.

Sucrose is a tasteless substance for most carnivorous fish species, but has an unpleasant taste for barramundi, as well as for kutum, *Rutilus frisii kutum*, and puffer, *Takifugu pardalis*, which belong to the same fish trophic category [50,56]. Undoubtedly, the deterrence of sucrose is related to the fact that barramundi feed exclusively on animals—fish and crustaceans [57]. Like all fish, barramundi have no specific requirement for dietary carbohydrates and have a limited capacity to digest different carbohydrate types [58]. In general, the barramundi's negative response to the taste of sucrose does not contradict the general rule that sucrose is palatable only for facultative and obligate herbivorous fish [27].

Barramundi manipulate the offered pellets by making repeated pellet rejections and re-grasps, over 20 times in some cases. More common are the cases with 1–2 grasps per trial (Figure 5). Such manipulations are typical of feeding behavior of many other fish species. The highest number of pellet grasps were observed in trials with pearl and three spots gouramis, and exceeded 30 in both species [26,28]. Barramundi make the least number of manipulations with the most attractive pellets flavored with cysteine, citric acid, and shrimp extract. This fully corresponds to the behavior of barramundi for the pellets containing highly palatable extracts of the sea urchin, *Diadema setosum* [20]. However, a significant relationship between palatability and the number of pellet grasps was not found (Figure 7).

Barramundi keep highly attractive pellets in the mouth for a longer period of time than they do with less attractive pellets. For example, pellets with cysteine and citric acid are retained after the first grasp 4.2 and 3.6 times longer than pellets with aversive tyrosine and sucrose, respectively. There is a strong positive relationship between the consumption of pellets and retention time after the first grasp and during the entire trial (Figure 7). This clearly shows that barramundi spend much more time on the orosensory evaluation of food before swallowing than before rejection. This pattern of feeding behavior is inherent in all previously studied fish [26,28], which gives us reason to believe that a long retention of food in the oral cavity before swallowing is a rule common to many, if not all, fish species.

In our experiments, we used relatively small pellets of about 5% of the length of the average fish body. However, many of the pellets were fragmented by the fish during testing. Pellets flavored with some substances were fragmented in almost half of the trials, 49.2 and 38.3% of the time for pellets with aspartic and glutamic acids, and 45% of the time for control pellets. It can be considered that the fragmentation of grasped pellets is typical feeding behavior of barramundi, and has been observed in previous experiments [20,21]. Fragmentation of pellets has been noted in trials on some other fish species as well, such as brown trout, as an example [39]. Barramundi are able to destroy pellets despite the fact that cutting and macerating teeth are absent in this fish species. In general, the pellet size is much smaller than the maximum size of prey consumed by fish in nature (60% of the consumer's body length) [59]. Yet, we might hypothesize that only juvenile barramundi, feeding mainly on small planktonic crustaceans, have this ability [57]. Large barramundi feeding on fish and macro-crustaceans in the wild may not have this ability and can swallow their food whole [59].

Pellets flavored with highly palatable cysteine, citric acid, and shrimp extract, as well as those with the most deterrent substances, including tyrosine isoleucine, are rarely destroyed. Therefore, the frequency of fragmentation of the pellets does not depend on palatability, nor on retention time in the oral cavity (Figure 7). Moreover, in many trials, the fish destroy the pellet very quickly, even after a single grasp, and after a very short time retaining it in the mouth. What causes such intra-oral food treatment, leading to the destruction of the pellets, remains unclear. It is possible that careful intra-oral processing leading to food fragmentation partly replaces the repeated re-grasp cycles in barramundi. It cannot be ruled out that such procedure of orosensory evaluation of food is associated with the morphology of taste buds and their distribution in the oral cavity of barramundi. To our knowledge, there are no such data in the literature, so the study of taste bud morphology and distribution in barramundi is interesting and promising.

It is obvious also that the ability of barramundi to expel and destroy pellets should be considered when new feeds are developed. Pellets used for barramundi feeding should be mechanically resistant so that feed loss owing to feed fragmentation will be minimal. In addition, this loss could be decreased if more palatable feeds are used for barramundi feeding.

Another feature of the feeding behavior of barramundi are the different patterns of pellet testing in trials that ended with swallowing or a final refusal to consume. Before swallowing barramundi keep pellets in the mouth several times longer than they do with pellets that they ultimately reject, and more often destroy them. This clearly shows that barramundi, like many other fish, spend much more time on orosensory evaluation of their food before swallowing [26]. It gives us reason to consider that a longer retention of food in the oral cavity before swallowing is a rule common to many, if not all, fish species. It would seem that fish need to subject their food to long and thorough orosensory control, in the case of swallowing, whereas they quickly refuse unsuitable food to save time in their search and evaluation of their next food. However, barramundi's ingestion of a pellet followed by a refusal to consume it is often preceded by nearly the same number of repeated re-grasp cycles, whereas many other fish perform fewer such cycles before swallowing the pellet than they do before refusing the pellet [26,27].

## 5. Conclusions

Our detailed analysis of taste responses in barramundi add to the convincing evidence that taste preferences are species-specific in fishes. The range of dissimilarity among amino acids in their palatability for various fish species creates an environment for selective feeding and decreases competition for food among sympatric fish species. The narrow spectra of attractive amino acids for barramundi shows once again that, despite their wide distribution in food organisms, amino acids, including essential ones, do not necessarily have to be attractive to fish. In contrast to fishes' widely-varying taste preferences, their feeding behaviors remain relatively similar between differing species. Barramundi, like most previously studied fish, manipulate food objects, making repeated cycles of grasps and rejections. As with all other fish studied, a clear relationship between the palatability of a food and the duration of its orosensory testing is characteristic of barramundi. The destruction of pellets is also a feature common to many fish species and is not exclusive to barramundi. Thus, the feeding behavior associated with orosensory testing of food is more similar in fish than taste preferences, which can evolve rapidly and provide adaptation of fish to new or shifted feeding conditions.

Currently, barramundi is farmed in many countries. The presented data about its specific taste preferences and feeding behavior can be implemented into applications for improving cultivation technology. Supplementation of feeding stimulants into commercial pelleted feeds formulated specifically for particular species is a promising method for promoting feed intake in fish [60,61]. Apparently, cysteine or citric acid alone is sufficient to function as an effective taste stimulant for the barramundi. However, mixtures that include both these and other substances, such as alanine, or another palatable substance can certainly become more effective. Screening of taste attractiveness in carboxylic acids can help us to find new taste stimulants. Another promising group of substances that may prove to be taste stimulants is fatty acids, which play a diverse role in the physiological and biochemical processes within aquatic animals [62,63]. Current evidence is emerging concerning fat as a sixth basic taste. Ongoing research also shows that fatty acids may cause strong taste sensations and improve fishes' food intake [64–66]. At the present moment, however, taste preferences of fatty acids for fish remain to be studied.

**Author Contributions:** Conceptualization, A.K.; Methodology, A.K. and O.I.; Validation, A.K.; Formal Analysis, L.T.K.O., O.I. and A.K.; Investigation, O.I.; Resources, L.T.K.O., O.I. and A.K.; Data Curation, O.I.; Writing—Original Draft Preparation, A.K.; Writing—Review and Editing, A.K., O.I. and L.T.K.O.; Visualization, L.T.K.O.; Supervision—A.K.; Project Administration, L.T.K.O.; Funding Acquisition, A.K. and, L.T.K.O. All authors have read and agreed to the published version of the manuscript.

**Funding:** This research was funded by the Russian Science Foundation, grant number 22-24-00125, https://rscf.ru/en/project/22-24-00125/ (accessed on 14 January 2022)., and by the Joint Vietnam–Russia Tropical Science and Technology Research Center, project "Ecolan E3.2, task 2".

**Institutional Review Board Statement:** The study was conducted in accordance with the Declaration of Helsinki, and the experimental protocol used in the present study was approved by the Joint Vietnam-Russia Tropical Science and Technology Research Center (permit # 856/QD-TTNDVN).

**Informed Consent Statement:** Not applicable.

**Data Availability Statement:** Not applicable.

**Acknowledgments:** The authors would like to thank the administration and staff of the Coastal Branch of the Joint Vietnam–Russia Tropical Science and Technology Research Center for their help in organizing field sample collection and kindly allowing us to use their laboratories and experimental facilities. Special thanks to the reviewers for their valuable contributions which helped improve the manuscript.

**Conflicts of Interest:** The authors declare no conflict of interest.

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
