# Peer review of "Taste Preferences and Orosensory Feed Testing Behavior in Barramundi Lates calcarifer (Latidae, Perciformes)"

_jmse, doi:10.3390/jmse10091213_

Round 1
Reviewer 1 Report
The manuscript reports on the taste preferences of a species of high aquaculture interest, expanding knowledge on gustatory preferences of a wide range of teleosts using a standard methodology which enables doing a comparative analysis of the spectra of responses across diverse fish species. The study is well performed and clearly reported, although it needs an extensive review by a native English speaker to bring it to a publication standard. A few suggestions and comments are made directly in the edited manuscript but these should not be taken as exhaustive and it is highly recommended that the manuscript undergoes an extensive language check. Some additional minor comments to be addressed are reported below:
- L110: how much is “an appropriate amount of water”? Considering that the water extract was used as such, it would be important to define this better, so that the experimental design is repeatable.
- L126: “several seconds” is too vague; please define a range of time. Furthermore, how can the authors be sure that there is no diffusion of water-soluble substances from the pellet to the water?
- L144: the test period from 9am to 17pm generates highly different conditions with respect to the fasting time and, hence, possibly hunger level. How did the authors have this factor into account, and can they comment if they have seen any difference in the response when a same substance was tested at different times – i.e., comparting the morning with the afternoon period?
- L192: Figure 2 legend – please note that standard error of the mean was previously defined as S.E. in the text (L153). Also, the same is observed in figures 3, 4, 6, 8, 9, 10.
- L197: Do you mean three times more (instead of better, which is a qualitative measure, not a quantitative one)?
- L198: “less” instead of “worse”? (same as previous comment).
- L201 and L204: Please revise the text and figure 3 – in the figure there are 6 amino acids with significant reduction and 13 amino acids without asterisks.
- Figure 5: This figure does not give additional information, and is redundant with the data shown in the table and described in the text. It could be removed or presented as a supplementary file.

Author Response
Thanks for your comments

Reviewer 2 Report
The paper “Taste preferences and orosensory feed testing behavior in barramundi Lates calcarifer (Latidae, Perciformes)” – ID: jmse-1858512 submitted to the special issue of JMSE Fisheries and Aquaculture: Current Situation and Future Perspectives contains interesting and important information on feeding behavior of a commercially valuable teleost fish. The obtained experimental data on the taste preference and the role of 21 free L-amino acids and 4 basic taste substances as taste stimulants are important and useful both in aquaculture and fish physiology/behavior. Within this wide range of substances, only three: citric acid, cysteine and alanine significantly enhanced taste attractivity of food pellets for barramundi juveniles; others were either aversive or inert. Authors thoroughly described and substantiated all aspects of experimental methods and design. The obtained data were adequately presented and illustrated with figures. Suitable statistical procedures were applied. Promising suggestions concerning the searching for other efficient taste stimulants and their combinations were suggested in the conclusions. Necessary literature on the main topic and biology of the studied species was cited. The text of the ms is clear and well structured. However, editing of the grammar and stylistic improvement are desirable. Below, some minor changes and corrections are listed, but some more efforts should be applied to improve the text.
The results were analyzed and discussed from different points of view, both practical and behavioral/physiological. To my opinion, the Discussion section is too large, mainly due to a sort of “meta-analysis” devoted to comparison of testing behavior of barramundi with that of more than 20 other fish species. Partly due to this, the percentage of self-citation is rather high (22 of 66 references). This part of the Discussion looks like a fragment of a special analytical study. I recommend to shorten it as much as possible.
Minor comments:
Lines:
L 24 - … in several times – delete “in”
L 62, 64 – attractivity, attractiveness – make consistent throughout the text
L 65 – colored instead of colorated
L 73-74 – not clear
L 82 – equipped instead of fitted
L 134 - … triggered well-visible chewing activity in fish
L 174 - … fear and hiding…; which is typical of…
L 203 – pronounced
L 207-208 – italic for Penaeus vannamei
L 213 – for the trials
L 207 – change flavoured for flavored throughout the text
L 346 – favorable; m.b. better stimulating?
L 367 – blends?
L 383 – typical of
L 389 – a significant relationship … was not found
L 390 – Barramundi keep highly attractive …
L 394-395 - …spend much more time… - more than what?
L 397 – long retention?
L 303 – typical of
L 407 - … is many times …
L 424 - grammar!
L 432-433 – check the sentences
L 429-440 – check it for repetition
L 460 – characteristic of
Author Response
Thanks for your comments.
